# Zinc Oxide Nanoparticles Induce Toxicity in H9c2 Rat Cardiomyoblasts

**DOI:** 10.3390/ijms232112940

**Published:** 2022-10-26

**Authors:** Criselda Mendoza-Milla, Fernanda Isabel Macías Macías, Kimberly Abigail Velázquez Delgado, Manuel Alejandro Herrera Rodríguez, Zaira Colín-Val, María del Pilar Ramos-Godinez, Agustina Cano-Martínez, Anita Vega-Miranda, Diana Xochiquetzal Robledo-Cadena, Norma Laura Delgado-Buenrostro, Yolanda Irasema Chirino, José Ocotlán Flores-Flores, Rebeca López-Marure

**Affiliations:** 1Laboratorio de Transducción de Señales, Unidad de Investigación, Instituto Nacional de Enfermedades Respiratorias Ismael Cosío Villegas, Ciudad de México 14080, Mexico; 2Departamento de Fisiología, Instituto Nacional de Cardiología Ignacio Chávez, Juan Badiano No. 1, Colonia Sección 16, Tlalpan, Ciudad de México 14080, Mexico; 3Departamento de Microscopía Electrónica, Instituto Nacional de Cancerología, Ciudad de México 14080, Mexico; 4Departamento de Bioquímica, Instituto Nacional de Cardiología Ignacio Chávez, Ciudad de México 14080, Mexico; 5Unidad de Biomedicina, Facultad de Estudios Superiores Iztacala, Universidad Nacional Autónoma de México, Tlalnepantla de Baz 54090, Mexico; 6Instituto de Ciencias Aplicadas y Tecnología, Universidad Nacional Autónoma de México, Ciudad de México 04510, Mexico

**Keywords:** zinc oxide nanoparticles, toxicity, cell death, cardiomyoblasts

## Abstract

Zinc oxide nanoparticles (ZnO NPs) are widely used in the cosmetic industry. They are nano-optical and nano-electrical devices, and their antimicrobial properties are applied in food packaging and medicine. ZnO NPs penetrate the body through inhalation, oral, and dermal exposure and spread through circulation to various systems and organs. Since the cardiovascular system is one of the most vulnerable systems, in this work, we studied ZnO NPs toxicity in H9c2 rat cardiomyoblasts. Cardiac cells were exposed to different concentrations of ZnO NPs, and then the morphology, proliferation, viability, mitochondrial membrane potential (ΔΨm), redox state, and protein expression were measured. Transmission electron microscopy (TEM) and hematoxylin–eosin (HE) staining showed strong morphological damage. ZnO NPs were not observed inside cells, suggesting that Zn^2+^ ions were internalized, causing the damage. ZnO NPs strongly inhibited cell proliferation and MTT reduction at 10 and 20 μg/cm^2^ after 72 h of treatment. ZnO NPs at 20 μg/cm^2^ elevated DCF fluorescence, indicating alterations in the cellular redox state associated with changes in ΔΨm and cell death. ZnO NPs also reduced the intracellular expression of troponin I and atrial natriuretic peptide. ZnO NPs are toxic for cardiac cells; therefore, consumption of products containing them could cause heart damage and the development of cardiovascular diseases.

## 1. Introduction

Zinc oxide nanoparticles (ZnO NPs) constitute a white thermostable material and are the most widely manufactured nanoparticles for personal care products including pigments, coatings, sensors, antibacterial creams, electronic gadgets, catalysts, food additives, and biomedical applications, among others [1,2]. Due to their small size (<100 nm), ZnO NPs can enter the human body through inhalation, ingestion, and dermal contact. Inhalation is the primary route in the occupational environment [3]. Dermal exposure happens through skin contact with cosmetic products [4], and oral exposure occurs through water and food containing ZnO NPs that were either released into the environment or used as additives [5]. After exposure, ZnO NPs spread through systemic circulation into multiple tissues and organs. Nanoparticles have been detected in the liver, spleen, lungs, kidneys, and heart [6]. In vivo and in vitro studies have shown several toxic effects of ZnO NPs, including zinc ion release, the generation of reactive oxygen species (ROS), the mechanical damage of cells, cytotoxicity, genotoxicity, and neurotoxicity as well as developmental toxicity [2]. 

Since the main route of nanoparticle distribution is through systemic circulation, accounting for the accumulation of ZnO NPs in the heart, exposure to this material has been widely correlated with cardiovascular diseases [7]. Oral administration of ZnO NPs in rats had cardiotoxic effects, increasing cardiac injury markers in serum, such as troponin-T, creatine kinase-MB, and myoglobin, and pro-inflammatory markers, such as tumor necrosis factor-α, interleukin-6, and C-reactive protein [8]. ZnO NPs also elevated cardiac calcium concentration, induced DNA oxidation, and stimulated caspase-3 activity [8]. In an in vitro study, ZnO NPs caused the dysfunction of human cardiac microvascular endothelial cells, increasing membrane permeability and inflammation [9]. Because there are few studies related to the effects of ZnO NPs on the heart, the underlying mechanisms remain unknown. Further studies are needed to confirm ZnO NPs cytotoxicity in cardiac cells. Here, we studied the cellular internalization of ZnO NPs and their impact on the morphology, proliferation, redox state, and viability of H9c2 rat cardiomyoblasts.

## 2. Results

### 2.1. ZnO NPs Characterization

Table 1 summarizes the physicochemical characteristics of ZnO NPs. Zetasizer determinations showed differences depending on the suspension medium. When ZnO NPs were suspended in DMEM plus fetal bovine serum (FBS), a smaller hydrodynamic size (350.4 ± 28.9 nm) was observed compared with ZnO NPs suspended in NaCl-HEPES buffer (NHB) (448.8 ± 42 nm) (Table 1). The zeta potential was −20.93 ± 0.25 mV for nanoparticles suspended in the culture medium and −4.30 ± 1.15 mV for those suspended in NHB. No change in the polydispersity index was obtained when comparing both suspension media.

The results obtained from the X-ray Powder Diffraction (XRD) pattern showed that the crystalline phases correspond to zinc oxide, according to the Joint Committee on Powder Diffraction Standards (JCPDS) card 01-080-7099 (Figure 1). On the other hand, the analysis of ZnO NPs by TEM supported estimates that the average crystal size is on the order of 10–50 nm (Figure 2).

### 2.2. ZnO NPs Were Not Observed Inside Cells

H9c2 cells were exposed to different concentrations of ZnO NPs for 24 h, and cellular uptake was evaluated by TEM. Cellular structures, such as the nucleus, nucleoli, mitochondria, and endoplasmic reticulum, showed no alterations in nonexposed cells. Nanoparticles were not observed inside the cells; however, important morphological changes were perceived at 5 μg/cm^2^ (Figure 3). Multiple vacuole formations were detected at 5 μg/cm^2^. At higher concentrations (≥ 10 μg/cm^2^), numerous large vesicles containing dense material were observed. Organelle disintegration was detected in the mitochondria and endoplasmic reticulum as well as the disruption of the cytoplasmic membrane and collapsed nuclei with a low density of nuclear material. These results indicated that ZnO NPs induced strong cellular damage. 

### 2.3. ZnO NPs Altered Cell Morphology

ZnO NPs-derived changes in cellular structure were identified by hematoxylin–eosin (HE) staining. Nonexposed cells displayed the fusiform morphology of epithelial appearance with elongated and abundant cytoplasm, central and round nuclei, and several mitoses (Figure 4). An important reduction in cell number was observed at 5 μg/cm^2^ of ZnO NPs. The most significant morphological changes were obtained at 20 μg/cm^2^, including compact nuclei, cytoplasm with limited stellar extensions, a plasma membrane rupture, and cytoplasm leakage, indicating strong cellular damage.

### 2.4. ZnO NPs Decreased Cell Proliferation and MTT Reduction

Since ZnO NPs induced strong morphological changes in the cells, which could be associated with the inhibition of cell proliferation or loss of viability, these events were determined by crystal violet and MTT reduction, respectively. An important decline in cell proliferation was observed at ≥10 μg/cm^2^ of ZnO NPs after 48 h of treatment, with the maximum effect (>50% of inhibition) at 72 h (Figure 5A). A lower MTT reduction of about 50% was obtained after 72 h of exposure at 10 and 20 μg/cm^2^ (Figure 5B).

### 2.5. ZnO NPs Altered Cellular Redox State 

Since ZnO NPs decreased cell proliferation and MTT reduction, the changes in the cellular redox state were examined (Figure 6). After 1 h of treatment with all the used concentrations of ZnO NPs, DCF fluorescence was increased, indicating an augmentation of reactive oxygen species (ROS) production and an alteration in the cellular redox state. The mean fluorescence intensity (MFI) of DCF increased in a concentration-dependent manner. The maximal value was observed at 10 µg/cm^2^, and the difference was significantly different from the control. When longer exposure times were used (3, 6, and 24 h), the cell damage was so strong that the cellular redox state could not be determined.

### 2.6. ZnO NPs Did Not Induce Mitophagy but Affected Mitochondrial Integrity 

ZnO NPs did not increase LTR-lysosome levels, indicating that mitophagy did not occur, but they severely affected nuclear and mitochondrial integrity through the loss of Hoechst and MitoTracker stainings, respectively, at 10 and 20 µg/cm^2^ (Figure 7). Since MitoTracker was lost during cell death due to the collapse by ΔΨm depolarization, this result could indicate a lower mitochondrial function which might compromise cell viability.

The ΔΨm status was verified using the Rh123 fluorescent probe (Figure 8). ZnO NPs significantly reduced fluorescence intensity by 33% and 51% at 10 and 20 μg/cm^2^, respectively, after 1 h of exposure, indicating a lower mitochondrial transmembrane inner potential and subsequent mitochondrial damage. Taken together, these results indicated that other mechanisms associated with dysfunctional oxidative phosphorylation may be involved in cardiomyocyte death.

### 2.7. ZnO NPs Induced Apoptotic and Necrotic Death

To evaluate whether the morphological changes were induced by ZnO NPs, as inhibition of cellular proliferation and metabolic activity as well as mitochondrial damage were associated with cell death, apoptosis and necrosis were quantified by annexin-V and propidium iodide staining. Control cells presented with about 5–10% death at baseline. ZnO NPs increased the levels of apoptotic cells in a concentration-dependent manner at all concentrations which was measured by fluorescence microscopy after 24 and 48 h, with the maximum effect at 48 h (Figure 9A,B). The necrotic cell number increased at 5 μg/cm^2^. The 20 μg/cm^2^ concentration was the most effective at inducing death, with 65.3% of cells being apoptotic and 28.8% being necrotic (Figure 9B). Flow cytometric analysis showed that ZnO NPs increased apoptotic death by 7-fold at 10 and 20 μg/cm^2^ after 24 h of treatment compared with untreated cells (Figure 9C). After 48 h of exposure at 5 μg/cm^2^, an 8-fold increase in apoptosis was detected.

### 2.8. ZnO NPs Inhibited Cardiac Protein Expression

Since ZnO NPs caused strong damage to cardiac cells, the expression of the biomarkers of cardiac muscle damage, such as troponin I and atrial natriuretic peptide, were evaluated. ZnO NPs (10 μg/cm^2^) significantly reduced intracellular troponin I and atrial natriuretic peptide expression by 27% and 20%, respectively (Figure 10). It is important to mention that 20 μg/cm^2^ of ZnO NPs were so damaging to cells that the total amount of protein was not enough to perform the determination of the levels of these proteins.

## 3. Discussion

ZnO NPs enter the body through dermal, oral, and inhalation routes. Then, due to their nanometric size, NPs spread through systemic circulation, reaching different organs and cell types. The potential toxicity of ZnO NPs has become an issue of debate given the increasing trend of using them in the manufacture of products for human consumption. The cardiovascular system is one of the most vulnerable systems. Therefore, this work evaluated the toxicity of ZnO NPs in H9c2 rat cardiomyoblasts. The hypertrophic response in these cells is similar to that of primary cardiomyocytes, thus being a suitable model for studying cardiac damage [10]. 

Our first approach was to characterize the ZnO NPs. The results obtained from the XRD pattern showed that ZnO NPs correspond to zinc oxide. According to the JCPDS card, the ZnO NPs used in this work were a pure material (Figure 1). The nanoparticle size described by the supplier (<50 nm) corresponded to the average size obtained by TEM (10–50 nm) (Figure 2). 

In order to determine ZnO NPs behavior in different suspension media, we evaluated the physicochemical characteristics of NPs dispersed in culture medium plus a serum and saline solution. Results showed that ZnO NPs dispersed in a cell culture medium supplemented with FBS displayed smaller hydrodynamic and agglomerate sizes compared with NPs suspended in NHB (Table 1). Since the stability of the nanoparticles in a liquid solution is crucial as it determines the sedimentation and agglomeration that occurs, our results demonstrated that ZnO NPs are more stable when suspended in a medium plus a serum. Agglomerate size is influenced by the components of suspension media [11,12]. The serum decreases the agglomerate size, preventing high aggregation due to steric stabilization and improving the dispersion stability of NPs [13]. We also determined the zeta potential of ZnO NPs, a key parameter for measuring the repulsive electrostatic forces between particles and that indicates the dispersion stability of the nanofluid. The higher the zeta potential, the longer the range of repulsive force; therefore, suspension stability is enhanced by increased zeta potential [14]. We found that ZnO NPs that were dispersed in cell culture media plus FBS displayed a zeta potential of −20.93 ± 0.25 mV, while ZnO NPs dispersed in NHB scored −4.30 ± 1.16 mV (Table 1), indicating a higher stability in ZnO NPs dispersed in a medium plus a serum. The literature shows that dissolutions of NPs with a zeta potential from −10 to +10 mV display neutral charges and low interaction with other molecules dispersed in the media [15]. In addition, zeta potentials from −30 to +30 mV encompass NPs agglomerates with a negative charge that make them able to interact with positively charged molecules, and this may be the case for ZnO NPs dispersed in the cell culture medium. The zeta potential of ZnO NPs was strongly influenced by FBS. This medium with a serum makes it protein-rich, increasing the dispersion quality of NPs suspensions through steric effects [14]. ZnO NPs probably interact with proteins through electrostatic attraction. It has been shown that plasma and blood proteins, such as albumin, immunoglobulin, and fibrinogen, bind gold nanoparticles [16]. The link between proteins and NPs enhances colloidal stability since proteins adsorbed on nanoparticle surfaces act like spring structures exerting repulsive forces and preventing contact between particles, thereby increasing colloidal stability. Smaller aggregates are more readily taken up by cells. This could partially explain ZnO NPs toxicity after uptake by H9c2 cells where NPs are dispersed in a medium containing a serum and where agglomerates may interact with positively charged biomolecules, causing the damage and disruption of cellular components. Other factors underlying NPs toxicity include concentration, surface charge, and particle size, among others.

In spite of the fact that ZnO NPs agglomerates dispersed in the culture medium were smaller than those dispersed in saline solution, TEM analysis did not show NPs inside cells; however, strong morphological damage to the whole cell, dependent on concentration, was detected (Figure 3 and Figure 4). We previously showed that food-grade titanium dioxide E171 alters F-actin distribution in H9c2 cells [17]. Actin, the major cytoskeletal protein in most cells, is responsible for mechanical support and determines cell shape and movement. ZnO NPs may alter actin filaments, causing morphological changes in membranes and intracellular structures, mitochondrial injury, and the outflow of organelles [18,19,20]. 

There are two ways for ZnO NPs entry into cells: 1) internalization as whole NPs and 2) internalization as free Zn^2+^ ions [2]. Since ZnO NPs were not detected inside H9c2 cells, we hypothesize that free Zn^2+^ ion uptake may underlie the cellular toxicity of ZnO NPs. Many studies illustrate ZnO NPs fragmentation releasing Zn^2+^ ions into extracellular fluids, which diffuse passively across the plasma membrane, as a critical step for cell toxicity [21,22]. One study on the mouse macrophage Ana-1 showed that dissolved Zn^2+^ ions play a key role in ZnO NPs toxicity [23]. Likewise, the antimicrobial activities of ZnO NPs, evaluated through the bioluminescence of *Photobacterium phosphoreum,* were due solely to released Zn^2+^ ions [24]. These results highlight Zn^2+^ ions as the main factor for ZnO NPs toxicity in H9c2 cells.

Morphological alterations induced by ZnO NPs were linked to decreased cell proliferation and MTT reduction (Figure 5), oxidative stress (Figure 6), mitochondrial dysfunction (Figure 7 and Figure 8), and cell death (Figure 9). Similar outcomes were derived from other investigations on normal cells. In RAW 246.7 macrophages, ZnO NPs elevated intracellular Zn^2+^ ion concentration, generating intracellular ROS, plasma membrane leakage, mitochondrial dysfunction, and cell death [25]. In mouse-derived spermatogonia (GC-1 spg cells), ZnO NPs induced apoptosis and autophagy through oxidative stress [26].

Oxidative stress has been described as a key component of ZnO NPs toxicity [27]. ZnO NPs entry into cells triggers a defense mechanism that generates ROS, surpassing the antioxidant systems and eventually leading to inflammation [28]. Inflammation affects the mitochondrial electron transport chain at the internal membrane, and it injures membrane and cellular components such as DNA, ultimately causing cell lysis and death [29,30,31,32,33]. Another mechanism involved in ZnO NPs toxicity is endoplasmic reticulum (ER) stress. In HUVEC cells, ZnO NPs triggered ER stress responses, followed by apoptosis, suggesting that ER stress can be an earlier endpoint for ZnO NPs toxicity [34]. We observed the very early toxic effects of ZnO NPs on H9c2 cells, indicating that oxidative stress could be involved.

We did not detect mitophagy after ZnO NPs exposure (24 h); however, severe damage overcame cellular defenses, resulting in nuclear and mitochondrial alterations and causing apoptosis and necrosis. Interestingly, our results in fluorescence microscopy revealed apoptosis and necrosis in exposed cells, whereas flow cytometry detected only apoptosis (Figure 9). The difference between these techniques is that flow cytometric analysis relies on cell suspensions. Necrotic cells are washed away during trypsinization, therefore passing by undetected. Therefore, fluorescence microscopy is the best choice to evaluate both apoptosis and necrosis.

Markers of cardiac injury include the natriuretic peptides released from myocytes in cardiac atria and ventricles in response to increased wall tension and stretch to reduce blood pressure, cardiac hypertrophy, and ventricular fibrosis [35,36]. On the other hand, cardiac troponin I is a structural peptide released after cardiomyocyte damage [37]. Therefore, we measured the levels of troponin I and atrial natriuretic peptide proteins in H9c2 cells exposed to ZnO NPs (Figure 10). Both total protein expressions dropped at 10 µg/cm^2^, thus we suppose that intracellular levels decreased while secreted proteins increased. This result supports the strong injury caused by ZnO NPs in cardiac cells. 

Remarkably, the ZnO NPs concentrations of 10 and 20 μg/cm^2^ (33 and 66 µg/mL) were the most toxic to H9c2 cells. A pharmacokinetic study of rats that were orally exposed to single-dose ZnO NPs showed that 50 mg/Kg ZnO NPs displayed the maximum plasma concentration which was near 100 µg/mL after 6 h [38]. Despite the concentrations tested in vitro differing from those in vivo, the ZnO NPs levels in our in vitro model are within the range of concentrations achieved in vivo during acute exposure. We hypothesize that the cellular damage exerted by ZnO NPs in organisms derives from the accumulation over time during chronic exposure. However, there is no conclusive data on the total urinary and fecal excretion of NPs [38].

Similar toxic effects induced by ZnO NPs in H9c2 cells have been described in vitro in metal-based NPs such as titanium dioxide (TiO_2_ NPs) and silver (AgNPs). Previous works using human endothelial cells, an excellent model for studying toxicity in the cardiovascular system, have compared the cytotoxicity of several metallic NPs, showing that ZnO NPs were more cytotoxic than Fe_3_O_4_, Al_2_O_3_, AgNPs, and TiO_2_ NPs [9,39,40]. Our work group previously showed that TiO_2_ NPs inhibit proliferation and induce apoptotic and necrotic death in human umbilical vein endothelial cells (HUVEC). Anyway, they provoked endothelial dysfunction and activation, mediating the release of inflammatory cytokines, the adhesion of monocytes, and the increased expression of adhesion molecules [41]. Similar results have been observed in ZnO NPs and AgNPs. In one study, ZnO NPs decreased cell viability and increased levels of 8-hydroxy-2’-deoxyguanosine (8-OHdG), interleukin (IL)-6, nitric oxide (NO), and downregulated cardiovascular disease-related genes in human coronary artery endothelial cells (HCAECs) [42]. On the other hand, AgNPs inhibited proliferation, damaged the cell membrane, and induced apoptosis. It also upregulated inflammatory cytokines, adhesion molecules, and the adhesion of monocytes to HUVEC [43]. The effect of TiO_2_ NPs and AgNPs has also been determined in cardiomyocytes. Rat and human cardiomyocytes exposed to 100 μg/mL and 10 μg/mL of TiO_2_ NPs reduced the contraction amplitude and the beating rate, respectively [44]. Exposure to TiO_2_ NPs has also been associated with arrhythmic events [45]. AgNP exposure exerts fleeting toxic effects on myocardial electrophysiology without affecting ROS production or membrane integrity [46]. To our knowledge, our results are the first evidence of damage induced by ZnO NPs in cardiac cells. These results suggest that the direct contact of metal NPs with endothelial and cardiac cells can be toxic and dangerous to health.

## 4. Methods and Materials

### 4.1. Materials 

Cell culture reagents including Dulbecco’s modified Eagle’s medium (DMEM) high glucose, Fetal Bovine Serum (FBS), antibiotic–antimycotic solution (Anti-Anti 100X), and 0.25% trypsin-EDTA solution were purchased from Thermo Fisher Scientific (Waltham, MA, USA). Western blot reagents were acquired from Bio-Rad (Hercules, CA, USA). Antibodies against β-actin, troponin I, and atrial natriuretic peptide were purchased from Santa Cruz Biotechnology (Dallas, TX, USA). The autophagy assay kit was supplied by ABCAM (Cambridge, UK). Annexin-V-FLUOS staining kit was provided by Roche (Nonnenwald, Germany). Annexin-V-FITC was provided by Biolegend (San Diego, CA, USA), and Annexin-V buffer and Propidium Iodine were supplied by BD (New Jersey, USA). Sterile plastic material for tissue culture and 8-well tissue culture chambers were obtained from Sarstedt (Nümbrecht, Germany). MitoTracker Green FM and LysoTracker were from Invitrogen (Carlsbad, CA, USA). ZnO nanopowder < 50 nm particle size (BET), >97% (cat. No. 677450), and all other reagents were purchased from Sigma-Aldrich (St. Louis, MO, USA). 

### 4.2. Cell Culture 

Cells derived from embryonic rat ventricular tissue (H9c2) were obtained from the American Type Culture Collection (ATCC^®^ CRL-1446). Cells were cultured in DMEM high glucose supplemented with 10% FBS and Anti–Anti solution (1×). Cells were maintained at 37 °C under a humidified atmosphere with 5% CO_2_.

### 4.3. ZnO NPs Characterization

For the hydrodynamic diameter, zeta potential, and polydispersity index measurements, ZnO NPs powder was suspended both in DMEM medium supplemented with 10% FBS and in 0.9% NaCl-HEPES buffer (NHB). Both suspensions were vortexed at 60 Hz for 10 min for subsequent reading on the Zetasizer Nano-ZS90 equipment. To identify the phases of ZnO NPs crystals, an XRD pattern was performed. For this, a Siemens D5000 diffractometer was used with Cu Kα radiation (λ = 1.541874 Angstrom) in the range of 20–90° in 2-Theta at a scanning step of 0.020°/1.2 s at room temperature. ZnO NPs were also analyzed by TEM to determine the nanoparticle size. NPs were examined in a JEOL transmission electron microscope model ARM200F, using copper Agar brand grids with formvar cover and carbon type A as well as with an acceleration voltage of 200 kV. 

### 4.4. ZnO NPs Internalization 

ZnO NPs internalization was detected by TEM. Cells were cultured with ZnO NPs (2.5, 5, 10, and 20 μg/cm^2^) for 24 h. After treatment, H9c2 cells were fixed with 2.5% glutaraldehyde-paraformaldehyde in phosphate buffer solution (PBS) (pH 7.2) for 45 min. Afterwards, fixation cells were placed in 1% osmium tetroxide for 1 h. Then, cells were dehydrated through graded series of alcohols and embedded in Epon 812 epoxy resin. Then, 60-nm-thin sections were obtained with a diamond knife on Ultracut-R ultramicrotome; mounted on copper grids; and impregnated with heavy metals, lead nitrate, and uranyl acetate. Grids were examined in a JEOL 10/10 transmission electron microscope at 60 kV and an AMT Camera System. 

### 4.5. Cell Morphology

For morphological analysis, H9c2 cells (20 × 10^3^) were seeded on 8-well tissue culture chambers; exposed to 2.5, 5, 10, and 20 μg/cm^2^ over 48 h; stained with hematoxylin–eosin (HE); analyzed with a microscope Olympus BX51 with 20X objective; and photographed with the camera Q Imaging MicroPublisher 5.0 RTV.

### 4.6. Cell Proliferation and Viability

Proliferation and viability were evaluated by crystal violet staining and MTT reduction respectively, as previously described by our work group [47]. H9c2 cells (3 × 10^3^/well) were cultured in 96-well plates and exposed to different concentrations of ZnO NPs (2.5, 5, 10, and 20 μg/cm^2^) over 24, 48, and 72 h. No exposed cells were used as a negative control. After exposure, cells were fixed with 1.1% glutaraldehyde for 10 min, washed twice with water, and stained with 0.1% crystal violet solution for 20 min. Incorporated crystal violet was solubilized in 100 μL/well of acetic acid (10%), and optical density at 590 nm was measured in a microplate spectrophotometer (Benchmark Plus, BIO-RAD). To assay viability, 20 μL/well of MTT (5 μg/mL) were added, and cells were incubated for 4 h at 37 °C. Then the medium was removed, formazan crystals were dissolved with acid isopropanol (0.04 N HCl), and optical density was read at 570 nm.

### 4.7. Oxidative Stress

The cellular redox state was measured by oxidation of 2′,7′-dichlorodihydrofluorescein diacetate (H_2_DCFDA) into 2,7-dichlorodihydrofluorescein (DCF) which is a redox indicator probe [48]. H9c2 cells (200 × 10^3^) were cultured with ZnO NPs (5, 10, 20 μg/cm^2^) for 1, 3, and 6 h in 56-mm glass Petri dishes. After treatment, cells were trypsinized and incubated with H_2_DCFDA (10 μM) for 30 min, and they were washed twice with PBS. Fluorescence was quantified in a FACSAria flow cytometer (Becton-Dickinson, CA, USA). 

Oxidative stress was also tested through ΔΨm changes based on the fluorescent dye rhodamine 123 (Rh123). For this, cell suspensions (1 × 10^6^) were treated with ZnO NPs (5, 10, 20 μg/cm^2^) for 1 h. Then, cells were washed with PBS and incubated with Rh123 (5 μg/mL) for 15 min. After incubation, cells were washed with PBS and analyzed in a FACSAria flow cytometer (Becton-Dickinson, CA, USA). Data was processed using FlowJo 8.7 software (Stanford University).

### 4.8. Mitophagy Assay

To assay mitophagy (selective removal of damaged mitochondria by autophagosomes and lysosomes), H9c2 rat cardiomyoblasts (50 × 10^4^) were cultured in 35-mm glass-bottomed Petri dishes (MatTek, Ashland, MA, USA) with ZnO NPs (2.5, 5, 10 and 20 μg/cm^2^) for 6 h. To detect nuclei, mitochondria, or lysosomes, cells were pre-incubated with 0.4 µM Bis-Benzimide H 33342 trihydrochloride (Hoechst), 0.5 µM MitoTracker Green (MTG), and 0.5 µM LysoTracker Red (LTR), respectively, for 30 min at 37 °C in DMEM without phenol red. Epifluorescence images were taken with the EVOS FL (Thermo Fisher Scientific, Waltham, MA, USA) cell-imaging microscope at 60X magnification.

### 4.9. Cell Death

The annexin-V-Fluos staining kit was used to determine cell death by microscopy. H9c2 cells (50 × 10^3^/well) were cultured in 6-well plates and treated with 2.5, 5, 10, and 20 μg/cm^2^ of ZnO NPs for 24 and 48 h. Afterwards, exposure cells were analyzed by fluorescence microscopy in a Floid Cell Imaging Station (Life Technologies, Carlsbad, CA, USA). For this, the culture medium was discarded, and cells were washed with PBS and subsequently incubated with 100 μL of annexin-V-Fluos labeling solution (20 μL propidium iodide plus 20 μL annexin-V-Fluos in 1 mL of incubation buffer) for 15 min then analyzed immediately. 

Apoptosis and necrosis were also examined through flow cytometry, using dual staining with annexin V-FITC and propidium iodide, according to the manufacturer’s instructions. Briefly, 1 × 10^6^ cells in 100 μL of annexin buffer were stained with propidium iodide staining solution and FITC annexin V staining solution. Cells were incubated at room temperature in the dark for 15 min and acquired in a FACSAria flow cytometer (Becton-Dickinson, CA, USA). For flow cytometry, cells were washed with PBS, trysinized, incubated with 100 μL of annexin-V-Fluos labeling solution, and immediately analyzed. Data was processed using FlowJo 8.7 software (Stanford University).

### 4.10. Protein Expression 

Western blot analysis was performed as described previously by our work group [49]. Cells were exposed to ZnO NPs (5, 10, 20 μg/cm^2^) for 48 h, and total proteins were isolated with a lysis extraction buffer. Proteins were quantified using Bio-Rad protein assay dye reagent concentrate. Proteins (30 μg) were separated through electrophoresis, using 8% SDS-polyacrylamide gels; transferred to PVDF membranes; blocked; and incubated with primary antibodies against troponin I, atrial natriuretic peptide, and β-actin as a load control (diluted 1:2500, 1:000, and 1:2500, respectively) overnight. Afterwards, membranes were washed three times with Tris Buffered Saline with Tween (TBS-T), incubated with the secondary antibody (diluted 1:2500) for 1 h, and washed again three times with TBS-T. Proteins were detected with the SuperSignal^®^ system and the ChemiDocTM MP Imaging System (Bio-Rad). Densitometric analysis for protein quantification was carried out with the Image Lab™ V 4.0 Software (Bio-Rad).

### 4.11. Statistical Analysis

To quantify and compare the patterns of actin structures, a nonparametric Kruskal–Wallis test was performed using the Prism 7.0a software (GraphPad software, Inc.). In order to determine statistical differences in all assays, Student’s *t*-tests were performed, and *p* < 0.05 was considered significant. For western blot analysis, multiple comparisons were based on one-way analysis of variance (ANOVA), followed by Turkey’s pairwise comparison as a post hoc test in Prism 5.01.

## 5. Conclusions

ZnO NPs induce severe damage to cardiomyocytes, characterized by lower cell proliferation, higher oxidative stress, cell death, and strong morphological changes with cellular disruption; therefore, the consumption of products containing ZnO NPs could cause the development of organ failure and cardiovascular diseases.

## Figures and Tables

**Figure 1 ijms-23-12940-f001:**
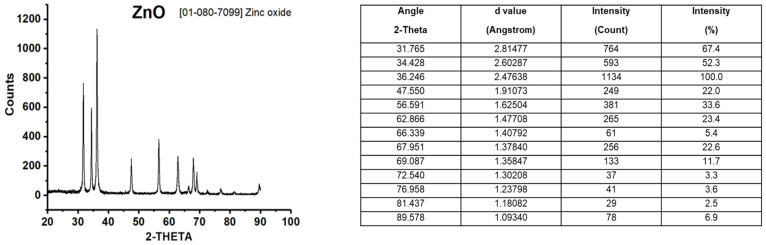
XRD pattern of ZnO NPs. NPs were analyzed in a diffractometer using Cu Kα radiation (λ = 1.541874 Angstrom) in the range of 20–90° in 2-Theta at a scanning step of 0.020°/1.2 s. The ZnO: JCPDS card [01-080-7099] is shown on the right side.

**Figure 2 ijms-23-12940-f002:**
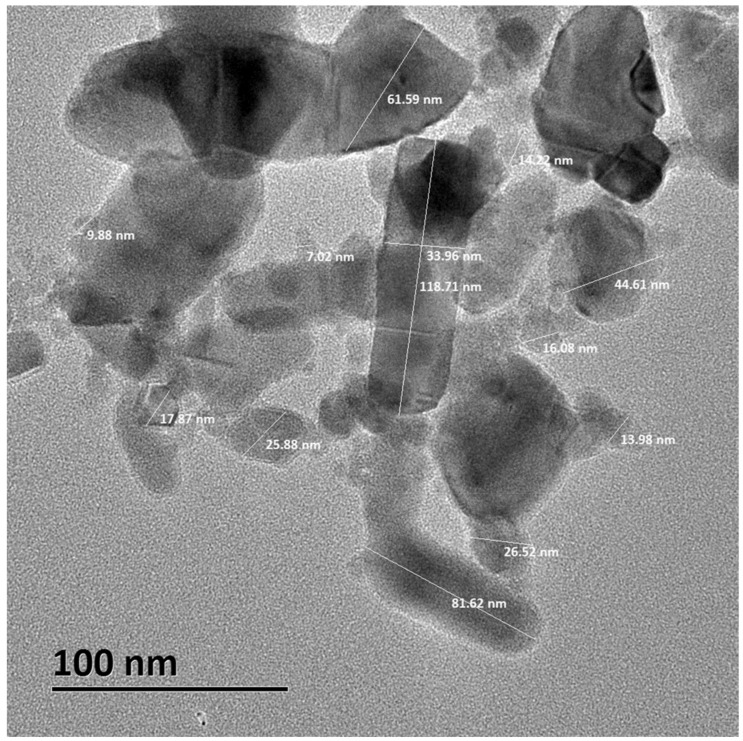
Size analysis of ZnO NPs by TEM. Microphotograph shows the different sizes of nanometric crystals in the nanopowder.

**Figure 3 ijms-23-12940-f003:**
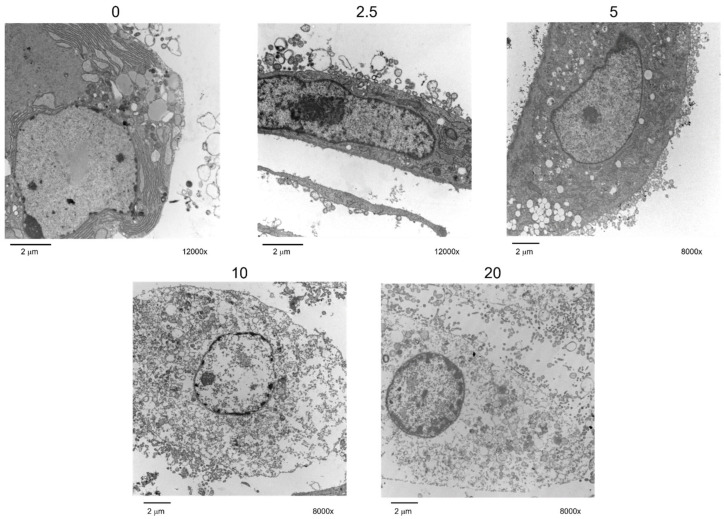
ZnO NP uptake by H9c2 cells. Cells (50 × 10^3^) were cultured in 6 well plates, exposed to 0, 2.5, 5, 10, and 20 μg/cm^2^ of ZnO NPs for 24 h, and they were analyzed by TEM with a JEOL 10-10 microscope and an ATM camera system. Images are shown at a direct magnification of 8000× and 12,000×. Bars = 2 μm. A representative image of three independent experiments is shown.

**Figure 4 ijms-23-12940-f004:**
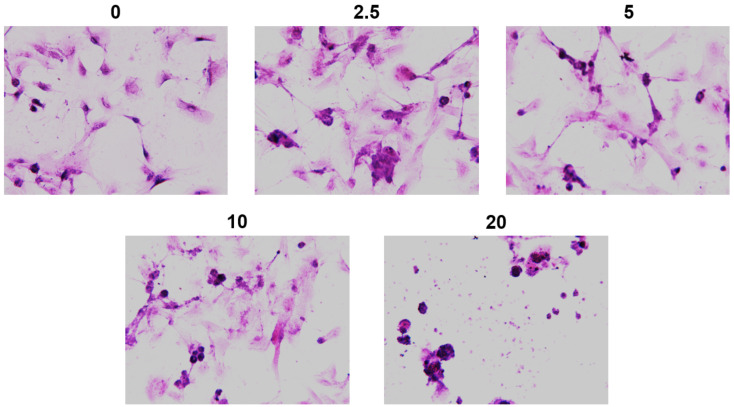
ZnO NP effects on cell morphology. Cells were exposed to 0, 2.5, 5, 10, and 20 μg/cm^2^ of ZnO NPs over 48 h. After treatment, cells were stained with hematoxylin–eosin (HE). Microphotographs were taken by an Olympus BX51 microscope at 20× magnification.

**Figure 5 ijms-23-12940-f005:**
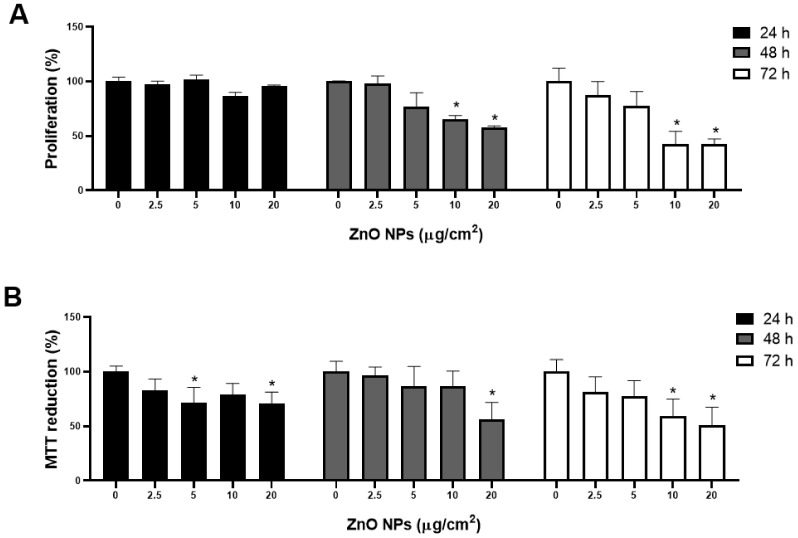
ZnO NP effects on cell proliferation and viability. Cells (3 × 10^3^/well) were exposed to 0, 2.5, 5, 10, and 20 μg/cm^2^ of ZnO NPs for 24, 48, and 72 h. Cell proliferation and viability were measured by crystal violet staining (**A**) and MTT reduction (**B**), respectively. Results were expressed as mean ± standard deviation of three independent experiments. * *p* < 0.05 compared with control.

**Figure 6 ijms-23-12940-f006:**
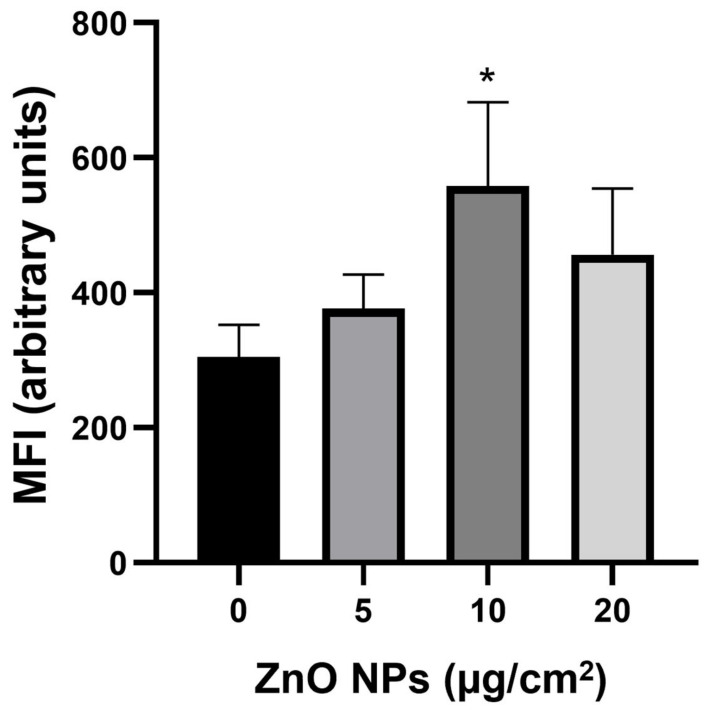
ZnO NPs effects on the cellular redox state. Cells (3 × 10^3^/well) were exposed to 0, 5, 10, and 20 μg/cm^2^ of ZnO NPs for 1 h. The cellular redox state was evaluated by H_2_DCFDA and analyzed by flow cytometry. Results were expressed as mean ± standard deviation of three independent experiments. MFI = mean intensity fluorescence. * *p* < 0.05 compared with control.

**Figure 7 ijms-23-12940-f007:**
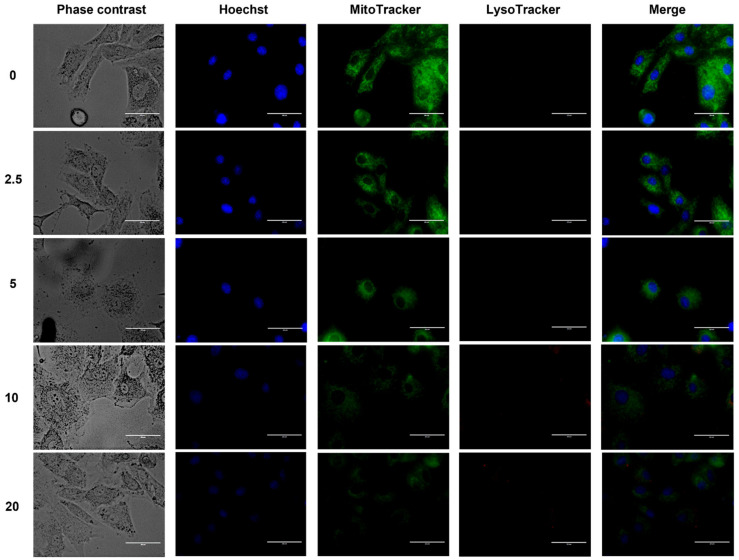
ZnO NPs effect on mitophagy. Cells were exposed to ZnO NPs (0, 2.5, 5, 10, and 20 μg/cm^2^) over 6 h. Cells were cultured in DMEM in glass-bottom culture dishes. Cell loading protocol was performed as indicated in the Material and Methods section. Bars = 50 μm.

**Figure 8 ijms-23-12940-f008:**
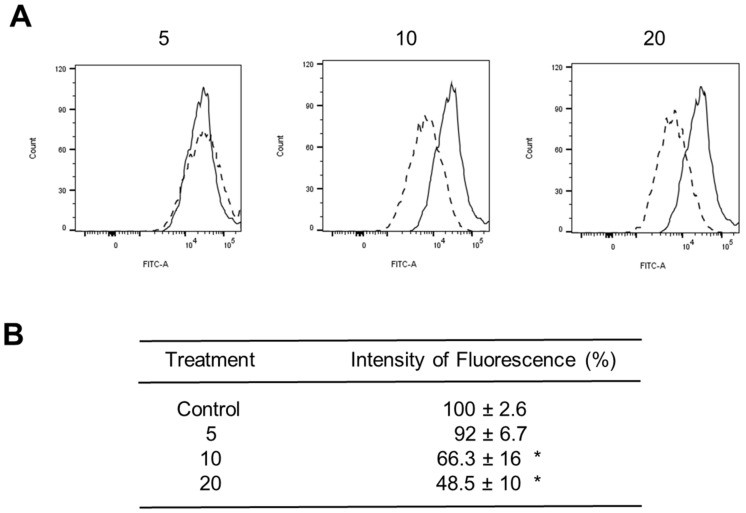
ZnO NPs effect on ΔΨm. Cells (1 × 10^6^) were exposed to 0 (Control), 5, 10, and 20 μg/cm^2^ of ZnO NPs for 1 h, and ΔΨm was measured using Rh123 and flow cytometry. In (**A**), a representative experiment is shown where the continuous line represents control cells, and the dashed line represents treated cells. In (**B**), results were expressed as mean ± standard deviation of three independent experiments with normalized vs. control cells set as 100%. * *p* < 0.05 compared with control.

**Figure 9 ijms-23-12940-f009:**
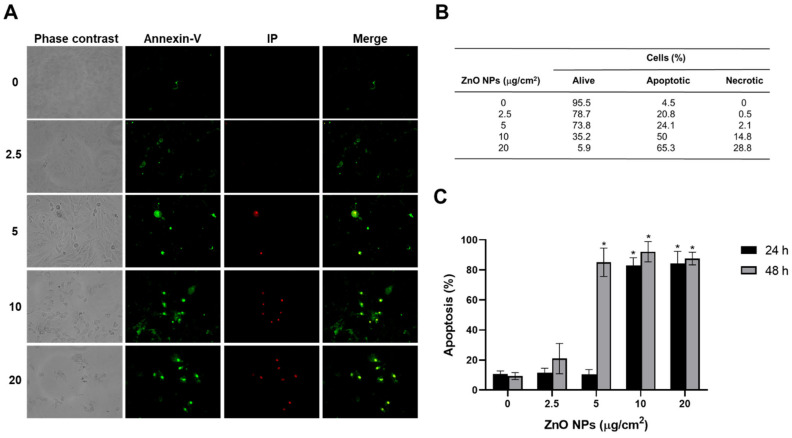
ZnO NPs effect on cell death. Apoptotic and necrotic cells were detected using annexin-V and propidium iodide, respectively. Cells (1 × 10^6^) were exposed to 0, 2.5, 5, 10, and 20 μg/cm^2^ of ZnO NPs. In (**A**), cell micrographs of a representative experiment after 48 h of exposure to ZnO NPs were taken by a fluorescence microscope with a 20X objective. In (**B**), the percentage of apoptotic and necrotic cells after 48 h of exposure is shown. In (**C**), flow cytometric analysis of cells is presented as mean ± standard deviation of three independent experiments. * *p* < 0.05 compared with control.

**Figure 10 ijms-23-12940-f010:**
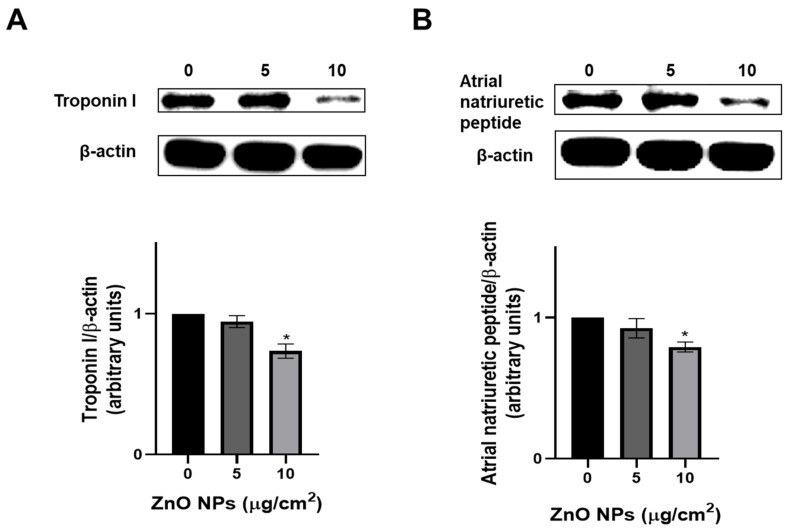
ZnO NPs effect on cardiac protein expression. Cells (1 × 10^6^) were exposed to 0, 5 and 10 μg/cm^2^ over 48 h, and the expression of troponin I (**A**) and atrial natriuretic peptide (**B**) was determined by western blot. A representative experiment is shown where β-actin was used as load control. Densitometric analysis of protein expression normalized against β -actin levels are showed below, and results are expressed as mean ± standard deviation of three independent experiments. * *p* < 0.05 compared with control.

**Table 1 ijms-23-12940-t001:** ZnO NPs physicochemical characteristics. ZnO NPs were suspended in DMEM plus 10% FBS and NHB at 20 μg/mL and were vortexed at 60 Hz for 10 min for subsequent reading on the Zetasizer Nano-ZS90 equipment.

Nanoparticles	Vehicle	Hydrodynamic Size (nm)	Zeta Potential (mV)	Polydispersity Index
ZnO NPs (20 µg/mL)	DMEM plus10% FBS	350.4 ± 28.9	−20.93 ± 0.25	1 ± 0.0
ZnO NPs (20 µg/mL)	NaCl-HEPES buffer	448.8 ± 42	−4.30 ± 1.16	0.96 ± 0.003

## Data Availability

All data generated or analyzed during this study are included in this published article.

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
