# Peer review of "Zinc Oxide Nanoparticles Induce Toxicity in H9c2 Rat Cardiomyoblasts"

_ijms, 2022, doi:10.3390/ijms232112940_

Round 1

Reviewer 1 Report

The authors report the toxicity of ZnO nanoparticles in H9c2 rat cardiomyoblasts. The reported issues have relevance in nanomedicine field. I believe this manuscript can be considered for publication in this Journal after major revisions.

-      The introduction part needs to update the information; please use the more up-to-date reference.

-      In the nanoparticle characterization section, provide an XRD pattern and SEM micrograph of ZnO NPs.

-      Nonsense lines were found in the article. Please correct it. (lines 269 to 273).

-      Figure 4 should be redesigned. (Fig. 4a) should be removed from my point of view, and the presented table should be displayed as a bar graph. (If possible, please use MFI index instead of positive report in ROS detection technique. To measure ROS by DCFHDA, it is better to use the MFI (Mean fluorescent intensity) index of flowcytometry. Cells are mainly positive for ROS status, but their intensity can differ.)

-      The authors should try comparing the toxicity properties of prepared particles with the literature.

-      The authors should try to give a comparison table to summarize the similar particles in the literature.

Author Response

Response to Reviewer 1 Comments

All changes made were highlighted in yellow.

  •     The introduction part needs to update the information; please use the more up-to-date reference.

Response 1: References for years below 2010 have been updated in the Introduction section.

-      In the nanoparticle characterization section, provide an XRD pattern and SEM micrograph of ZnO NPs.

Response 2:  The XRD pattern requested was performed to ZnO NPs. Analysis of nanoparticle size by SEM was not possible but analysis by TEM was performed. Compared to SEM, in which the electrons are mainly reflected from the surface to generate the image, in TEM the electrons penetrate through the studied sample providing more details, resulting in a more precise tool. The description of the used methods was added in the Material and Methods, and the description of these new results was added in the Results section. Two more figures were added with these results; therefore, the numbers of all figures were changed. Since the new results were obtained by another colleague, his name and ascription were added to the list of authors.

-      Nonsense lines were found in the article. Please correct it. (lines 269 to 273).

Response 3: The typographical errors and nonsense lines in page 12 were corrected.

-      Figure 4 should be redesigned. (Fig. 4a) should be removed from my point of view, and the presented table should be displayed as a bar graph. (If possible, please use MFI index instead of positive report in ROS detection technique. To measure ROS by DCFHDA, it is better to use the MFI (Mean fluorescent intensity) index of flow cytometry. Cells are mainly positive for ROS status, but their intensity can differ.)

Response 4:  Figure 4A (now figure 6) was eliminated and a new bar graph showing the mean fluorescence intensity in arbitrary units was added.

-      The authors should try comparing the toxicity properties of prepared particles with the literature.

Response 5:  A paragraph discussing the toxic effects induced by other metal based nanoparticles such as TiO2 NPs and AgNPs in models in vitro reported in the literature was added at the end of the Discussion section.

-      The authors should try to give a comparison table to summarize the similar particles in the literature.

Response 6:  Since a paragraph discussing the toxic effects induced by other metal NPs was added in the Discussion, we consider that the comparative table is no longer necessary. 

Reviewer 2 Report

In this manuscript, Mendoza-Milla et al, addressed the “Zinc oxide nanoparticles induced toxicity in H9c2 rat cardiomyoblasts”. Having examined the manuscript, I note that though it discusses some interesting observations, to be considered for International Journal of Molecular Sciences (ISSN 1422-0067), the following are some of the comments that the authors might find useful for future submission. This manuscript is addressing cardio toxic effects of zinc oxide nanoparticles This type of manuscript is extremely valuable at the community and global level.

General comments

11. The title is appropriately captured, and manuscript is well written.

  2. Authors need to correct the typographical errors observed in page 12, line numbers 270, 271, 272, and 273

33. Can H9c2 rat cardiomyoblasts accurately mimic the hypertrophic responses of human primary cardiac myocytes?

44. What is the stability of zinc oxide nanoparticles?

Author Response

Response to Reviewer 2 Comments

All changes made were highlighted in yellow.

  1. The title is appropriately captured, and the manuscript is well written.

Response 1:  Does not apply.

  1. Authors need to correct the typographical errors observed in page 12, line numbers 270, 271, 272, and 273

Response 2:  The typographical errors in page 12 were corrected.

  1. Can H9c2 rat cardiomyoblasts accurately mimic the hypertrophic responses of human primary cardiac myocytes?

Response 3: Yes. In the Discussion section, the last sentence of the first paragraph it is mentioned that H9c2 cells mimic the hypertrophic responses of neonatal rat primary cardiocytes, there is a very interesting article that demonstrates this: Watkins SJ, Borthwick GM, Arthur HM. The H9c2 cell line and primary neonatal cardiomyocyte cells show similar hypertrophic responses in vitro. In Vitro Cell Dev Biol Anim. 2011;47(2):125-31.

  1. What is the stability of zinc oxide nanoparticles?

Response 4:  The stability of nanoparticles refers to their behavior in suspension and determines whether sedimentation and aggregation occur. A smaller hydrodynamic size implies better stability and less aggregation.Dispersion stability of nanofluid can be measured through absolute value of zeta potential. ZnO NPs dispersed in medium containing serum had a zeta potential of -20.93 mV compared with the dispersed in NHB buffer (-4.3 mV), indicating a higher colloidal stability of dispersion. All these results were included in the Discussion section.

Round 2

Reviewer 1 Report

The authors report the toxicity of ZnO nanoparticles in H9c2 rat cardiomyoblasts. The reported issues have relevance in nanomedicine field. The results are interesting and suitable for publication. The experiments, results and discussion as well as supporting information are well presented and this paper is ready for publication in every respect.